# Improving Visual Quality of Image Synthesis by A Token-based Generator with Transformers

**Yanhong Zeng**[1,2*]   **Huan Yang**[3]   **Hongyang Chao**[1,2]   **Jianbo Wang**[4]   **Jianlong Fu**[3]

[1]School of Computer Science and Engineering, Sun Yat-sen University
[2]Key Laboratory of Machine Intelligence and Advanced Computing, Ministry of Education, China
[3]Microsoft Research Asia      [4]University of Tokyo

## Abstract

We present a new perspective of achieving image synthesis by viewing this task as a visual token generation problem. Different from existing paradigms that directly synthesize a full image from a single input (e.g., a latent code), the new formulation enables a flexible local manipulation for different image regions, which makes it possible to learn content-aware and fine-grained style control for image synthesis. Specifically, it takes as input a sequence of latent tokens to predict the visual tokens for synthesizing an image. Under this perspective, we propose a token-based generator (i.e.,TokenGAN). Particularly, the TokenGAN inputs two semantically different visual tokens, i.e., the learned constant content tokens and the style tokens from the latent space. Given a sequence of style tokens, the TokenGAN is able to control the image synthesis by assigning the styles to the content tokens by attention mechanism with a Transformer. We conduct extensive experiments and show that the proposed TokenGAN has achieved state-of-the-art results on several widely-used image synthesis benchmarks, including FFHQ and LSUN CHURCH with different resolutions. In particular, the generator is able to synthesize high-fidelity images with $1024 \times 1024$ size, dispensing with convolutions entirely.

## 1   Introduction

Unconditional image synthesis generates images from latent codes by adversarial training [10, 16, 19, 25, 27, 33, 50]. Recent advances have been achieved by a style-based generator architecture in terms of both the visual quality and resolution of generated images [26, 27, 28, 37, 52]. In particular, the style-based generator has been widely used in many other generative tasks, including facial editing [9, 40], style transfer [1, 38], image super-resolution [17, 31], and image inpainting [2, 52].

The key to the success of the style-based generator lies in the learning of the style control based on the intermediate latent space $\mathcal{W}$ [27, 28]. Instead of feeding the input latent code $\mathbf{z} \in \mathcal{Z}$ through the input layer only (Figure1-a), the style-based generator maps the input $\mathbf{z}$ to an intermediate latent space $\mathbf{w} \in \mathcal{W}$, which then controls the "style" of the image at each layer via adaptive instance normalization (AdaIN [21]) (Figure 1-b). It has been demonstrated that such a design allows a less entangled representation learning in $\mathcal{W}$, leading to better generative image modeling [12, 23, 27, 40].

Despite the promising results, the style-based generator can suffer from the style control via AdaIN operation [28, 52]. Specifically, the style control is content-independent. It "washes away" the original information of features by normalization and assigns new styles decided by the latent codes regardless of the image/feature content. Besides, the style code $\mathbf{w}$ affects the entire image by scaling and biasing complete feature maps with a single value via AdaIN operation [35, 44, 54]. Such an imposed single style over multiple image regions can inevitably result in entangled representation of

---

*This work was done while Yanhong Zeng was a research intern at Microsoft Research Asia.

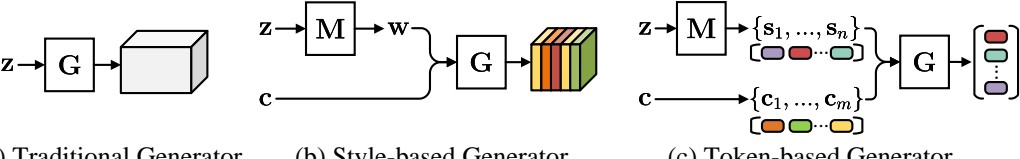

| (a) Traditional Generator | (b) Style-based Generator | (c) Token-based Generator |

Figure 1: Overview of different generators. **(a)** A traditional generator [16] feeds a single latent $\mathbf{z}$ as input to control the image synthesis. **(b)** A style-based generator [27] maps $\mathbf{z}$ to an intermediate latent space $\mathbf{w} \in \mathcal{W}$ to control the styles of the content $\mathbf{c}$ via AdaIN [21]. **(c)** Our token-based generator starts from a sequence of content tokens $\{\mathbf{c}_1, \cdots, \mathbf{c}_m\}$ and controls each content token with a set of style tokens $\{\mathbf{s}_1, \cdots, \mathbf{s}_n\} \in \mathcal{S}$ by attention mechanism with a visual Transformer. $\mathbf{M}$ denotes the mapping network and $\mathbf{G}$ denotes the generator network.

different image variations (e.g., hairstyle, facial expression) [28, 44, 52]. These limitations for image modeling can even lead to visible artifacts in the synthesized results (e.g., droplet artifacts [27]).

To get rid of the issues caused by StyleGAN's style modeling, we introduce a new perspective that views image synthesis as a visual token generation problem. The visual token is a popular representation of an image patch with a predefined size and position [6, 7, 13]; and has shown an impressive superiority in various tasks with the development of Transformer models [6, 13, 43, 46]. Inspired by the appealing property of the token-based representation, we propose to achieve image synthesis by visual token generation. Specifically, it takes as input a sequence of latent tokens to predict the visual tokens of an image. Such a token-based representation enables a flexible local manipulation for different image regions, which makes it possible to learn content-aware and fine-grained style control for image synthesis.

Under this new paradigm, we design a token-based generator, i.e., TokenGAN, for the visual token generation problem. Specifically, we consider two different types of input tokens in the generator, i.e., the content tokens and the style tokens. The content tokens are learned as the constant input in the generator network and the style tokens are projected from a learned intermediate latent space (Figure1-c). Given a sequence of style tokens, the TokenGAN learns to control the visual token generation by rendering each content token with related style tokens according to their semantics. In particular, since the Transformer has been verified to be effective in sequence modeling in a broad range of tasks [6, 43], we adopt a generator network architecture from a visual Transformer to model the relations between the content tokens and the style tokens. Through such a content-dependent style modeling by the attention mechanism in Transformer, the TokenGAN is able to achieve content-aware and fine-grained style learning for image synthesis.

We conduct both quantitative comparisons and qualitative analysis on several unconditional image generation benchmarks. The results show that the token-based generator has achieved comparable results to the state-of-the-art in image synthesis. We summarize our contributions as below:

- We propose a new perspective of achieving image synthesis by visual token generation. Such a token-based representation enables flexible local manipulation for different image regions, leading to a better image modeling.
- We propose a token-based generator (i.e., TokenGAN) for the visual token generation. Specifically, the TokenGAN introduces the style tokens and the content tokens. It adopts a Transformer-based network for content-aware style modeling.
- We show extensive experiments (quantitative and qualitative comparisons, study on style editing, image inversion and image interpolation) to verify the effectiveness of the token-based generator. Specifically, the token-based generator is able to synthesize high-fidelity $1024 \times 1024$ images without any convolutions in the generator.

## 2 Related works

### 2.1 Style-based generator

The distinguishing feature of the style-based generator is its unconventional generator architecture [26, 27, 28, 52]. Typically, the style-based generator consists of a *mapping network f* and a *synthesis*

*network g.* The mapping network is used to transform the input latent code $\mathbf{z}$ to an intermediate latent code $\mathbf{w} \in \mathcal{W}$ for learning a less entangled latent space. The synthesis network follows a progressive growing design that is able to first output low-resolution images that are not affected significantly by high-resolution layers [25, 27]. Such an architecture design can lead to an automatic, unsupervised separation of high-level attributes in different layers. For example, when trained on human faces, the low-resolution layers control coarse styles (e.g., pose, identity) of images and high-resolution layers control fine styles (e.g., micro-structure, color scheme) [25].

To control the attributes of the image, the style-based generator produces *styles* from the intermediate latent code $\mathbf{w}$ by affine transforms, which then control each layer of the synthesis network via adaptive instance normalization (AdaIN) [21, 35]. The AdaIN operation removes styles from previous layers by normalizing each feature map to zero mean and unit deviation, and it assigns the new *styles* by scaling and biasing the complete normalized feature maps. However, it has been demonstrated such an AdaIN operation can destroy the magnitude information of features and thus results in the well-known droplet artifacts [28]. Besides, the scale-specific style disentanglement in StyleGAN can be limited due to the complex semantic components in each feature map [28, 52]. In this paper, we explore a new architecture by flattening the image as a sequence of tokens, which enables fine-grained control of the image by assigning token-wise semantic-aware styles based on the attention mechanism.

## 2.2 Transformer in vision

The Transformer typically takes as input a sequence of vectors, called tokens [43]. The Transformer is built for sequence modeling solely on attention mechanisms over the tokens, dispensing with recurrence and convolutions entirely [5, 11, 43, 30]. Due to its great success in the field of natural language processing, an increasing number of works attempt to extend Transformer for computer vision tasks [6, 7, 8, 13, 18, 36, 55]. For example, iGPT shows promising results in image classification by pre-training a sequence Transformer with the task of auto-regressive next pixel prediction and masked pixel prediction [8]. Dosoyvitskiy et al. propose to split an image into patches and feed these patches into a standard Transformer (ViT) for image classification [13]. They show that a ViT with large-scale training can trump CNNs equipped with inductive bias. DETR is a seminal work that views object detection as a direct set prediction problem, eliminating the need for hand-crafted components and achieving impressive performance by Transformer [6]. To mitigate the issue of high computation complexity associated with long sequences caused by high-resolution images, a branch of works explores lightweight Transformer (e.g.., Deformable DETR) for vision tasks [55].

Transformer is also attracting increasing attention in low-level vision tasks [14, 24, 34, 36, 42]. Parmar et al. cast image generation as an autoregressive sequence generation problem and propose Image Transformer with a local self-attention mechanism [36]. However, Image Transformer can suffer from its quadratic computation cost and its long inference time due to auto-regressive prediction. Most existing works adopt a hybrid CNN-Transformer architecture, which consists of a CNN head for feature extraction, a Transformer encoder-decoder backbone, and a CNN tail for feature decoding [7, 46, 48]. For example, IPT fully utilizes the Transformer architecture by a large-scale pre-training and achieves promising results in several image restoration tasks [7]. In parallel work, Hudson et al. introduce bipartite structure to maintain computation of linear efficiency for long-range interactions across the image based on a convolution backbone [22]. Our token-based generator inherits the network architecture from Transformer without any convolution layers and is able to yield surprising promising results for high-resolution image synthesis.

## 3 Approach

In this section, we introduce the details of the proposed token-based generator (TokenGAN). As depicted in Figure 2, the TokenGAN takes as input two kinds of tokens to generate the visual tokens of images by a Transformer. We introduce the input tokens in Section 3.1 and the token-based generator architecture in Section 3.2, following the overall optimization in Section 3.3.

### 3.1 Visual tokens

It's intuitive that the visual tokens of an image contain the information of both content (i.e., semantics) and related styles. In our token-based generator, we choose to separate these two kinds of information

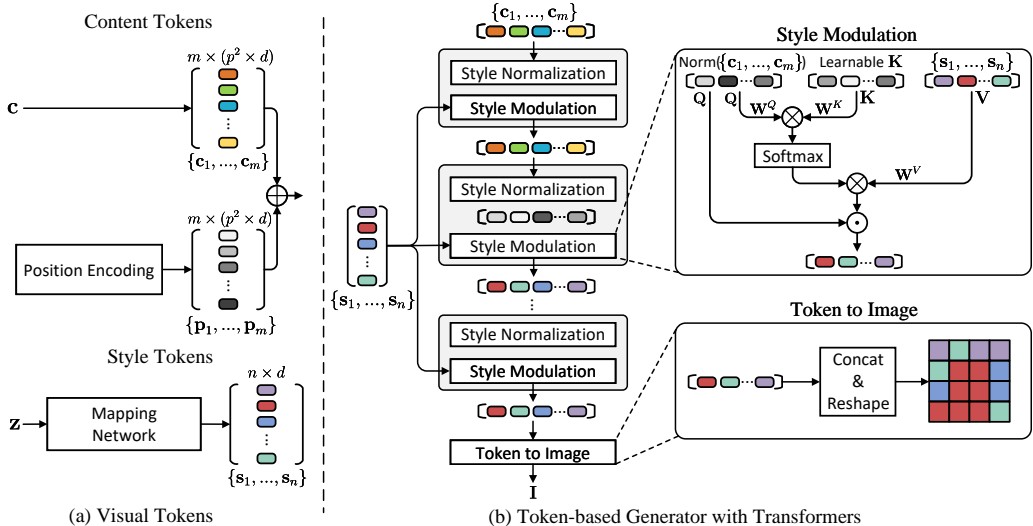

Content Tokens

Style Tokens

(a) Visual Tokens

(b) Token-based Generator with Transformers

Figure 2: The overview of the TokenGAN for the visual token generation task for image synthesis. The TokenGAN takes as input two kinds of visual tokens, i.e., the style tokens and the content tokens, to generate visual tokens of an image. Specifically, TokenGAN learns to render each content token by attended style tokens with a Transformer, leading to content-aware and fine-grained style control.

by a sequence of content tokens and a sequence of style tokens, so that the interactions between them can be modeled and controlled for the image synthesis. We introduce the details of each as below.

**Content tokens.** As depicted in Figure 2-a, the TokenGAN starts from the sequence of learned constant content tokens $\mathbf{c} \in \mathbb{R}^{m \times (p^2 \times d)}$ through the input layer. Specifically, $d$ is the dimension in terms of channels, $p \times p$ is the patch size of the content token, and $m$ is the length of the content token sequence. To maintain the position information of each token, we add position encodings $\mathbf{p} \in \mathbb{R}^{m \times (p^2 \times d)}$ to each content token following a commonly used paradigm [7, 13]. For simplicity, we rewrite the notation for the content tokens as $\{\mathbf{c}_1, \cdots, \mathbf{c}_m\}$, where $\mathbf{c}_i \in \mathbb{R}^d$.

**Style tokens.** Given a latent code $\mathbf{z}$ from the input latent space $\mathcal{Z}$, the mapping network adopts several MLPs to map the input $\mathbf{z}$ to a set of different style tokens $\{\mathbf{s}_1, \cdots, \mathbf{s}_n\} \in \mathcal{S}$, where $\mathbf{s}_i \in \mathbb{R}^d$. As shown in Figure 2-b, the style tokens are paired with a set of learnable semantic embedding as a key-value structure in each style modulation layer of the TokenGAN. With the Transformer modeling, all the content tokens will match with the semantic embedding and then fetch the new style from the style tokens based on the matching results. The fetched new styles are used to control the values of the content tokens, which are finally decoded to images. Such a token-wise style control enables content-aware and fine-grained style learning for image synthesis.

### 3.2 Token-based generator

As shown in Figure 2, the TokenGAN consists of multiple layers of style blocks and each style block consists of a style normalization layer and a style modulation layer. Specifically, the style normalization layer removes the styles from previous layers by relieving the dependence on the original statistics of input token features. At the same time, the style modulation layer assigns new styles from the current layer to the content tokens. In particular, the new styles are calculated based on the pairwise interactions between the style tokens and the content tokens by a cross-attention mechanism. We introduce more details about the style normalization, content-aware style modeling, and style modulation in our generator as below.

**Style normalization.** We apply style normalization on the input content tokens to remove the styles from previous layers, which is denoted as:

$$\text{Norm}(\mathbf{c}_i) = (\mathbf{c}_i - \mu(\mathbf{c}_i)) \,/\, \sigma(\mathbf{c}_i), \quad i = 1, \cdots, m, \tag{1}$$

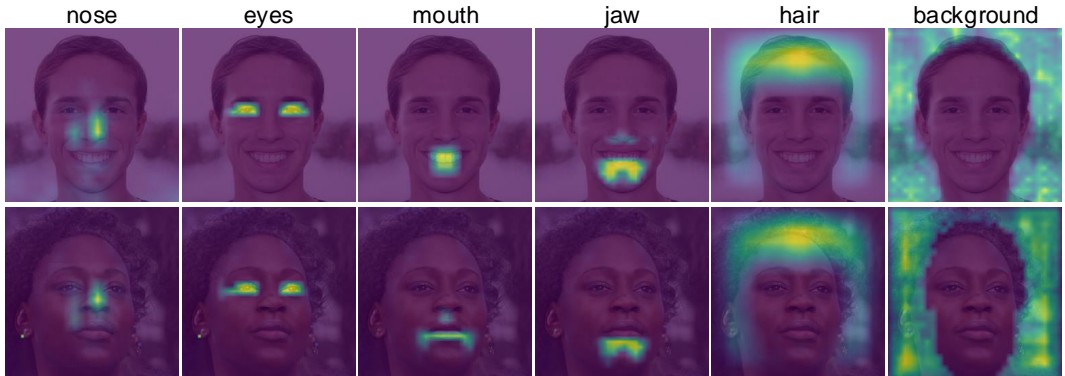

| nose | eyes | mouth | jaw | hair | background |

Figure 3: Visualization of the attention maps obtained by Eq. (2). Each column highlights the image regions that are affected most by the corresponding content-aware style token. It shows that TokenGAN is able to attend to meaningful semantics for different persons in an unsupervised way. The tags are associated by human based on the attention responses for clear presentation.

where each $\mathbf{c}_i \in \mathbb{R}^d$ denotes a content token, $d$ denotes the dimension of each content token, and $m$ is the length of the token sequence. Specifically, the style-based generator typically removes the styles from previous layers by *Instance Normalization* [41], which may destroy the information conveyed by the magnitude of the feature map relative to each other [27, 28, 41]. To avoid the above issue, we adopt *LayerNorm* [43] to relive the style dependence while maintaining the information of the relative magnitude following previous works [28, 29, 43].

**Content-aware style modeling.** After style normalization, the TokenGAN calculates new styles to control the input content tokens. As shown in the style modulation in Figure 2-b, the style tokens are paired with a set of learnable semantic embedding as a key-value structure. Given the content tokens $\{\mathbf{c}_1, \cdots, \mathbf{c}_m\}$ from the input and the style tokens from the mapping network $\{\mathbf{s}_1, \cdots, \mathbf{s}_n\} \in \mathcal{S}$, the token-wise new styles are calculated by attention mechanism:

$$\mathbf{S}' = Attention(\mathbf{C}, \mathbf{K}, \mathbf{S}) = softmax(\frac{\mathbf{C}\mathbf{K}^T}{\sqrt{d}})\mathbf{S}, \tag{2}$$

where $\mathbf{S}' \in \mathbb{R}^{m \times d}$ denotes new styles for all the content tokens, the input content tokens are packed together into a matrix $\mathbf{C} \in \mathbb{R}^{m \times d}$, similarly with the style tokens as matrix $\mathbf{S} \in \mathbb{R}^{n \times d}$, and the matrix $\mathbf{K} \in \mathbb{R}^{n \times d}$ for the learnable semantic embedding in where each row vector indicates a learned semantic. We omit the linear projection in the standard attention calculation for simplicity [43].

Through such a cross-attention mechanism, the new token-wise styles are fetched for style modulation based on the matching results. Particularly, tokens with similar semantics will have similar styles (e.g., the same color for the two eyes). We visualize the attended regions for different key-value pairs (i.e., semantic-style embedding) in Figure 3, and we can find that the learned embedding $\mathbf{K}$ is able to attend to meaningful image regions.

**Style modulation.** Different from the traditional Transformer that adds the attention results back to the input features as residual features, our attention results are used as new styles by amplifying each channel of the content tokens:

$$\mathbf{C}' = \mathbf{C} \odot \mathbf{S}', \tag{3}$$

where each row vector in $\mathbf{C} \in \mathbb{R}^{m \times d}$ indicates a content token, and $\odot$ indicates element-wise multiplication. Such a style modulation affects the operation of subsequent embedding layers (which we implemented by fully-connected layers) and thus control the style of generated images.

**Implementation details.** We translate the generated visual tokens to the image by concatenation and reshaping. In practice, we follow StyleGAN2 and adopt a skip-generator architecture in Token-GAN. Such a skip generator generates images in each layer at different resolutions (e.g., from $4^2$ to $8^2$) and progressively upsamples and sums the images from the previous layer by skip connections to the next layer [28, 25]. After progressive summing, the generator takes the output in the last layer as the final results.

## 3.3 Overall optimization

**Style mixing.** To further encourage the styles to localize, we employ the mixing regularization technique during training [27]. To be specific, a given percentage of images are generated using two random latent codes. We run two latent codes $\mathbf{z}_1$, $\mathbf{z}_2$ through the mapping network, and have the corresponding $\{\mathbf{s}_1^1, \cdots, \mathbf{s}_n^1\}$, $\{\mathbf{s}_1^2, \cdots, \mathbf{s}_n^2\}$. We randomly choose an inject point to mix the styles, so that a part of the final style tokens from $\mathbf{z}_1$ and another part from $\mathbf{z}_2$. This regularization technique prevents the network from assuming that a set of style tokens are correlated. A similar strategy is adopted in terms of different layers following the style-based generator [27].

**Optimization objectives.** We denote the generated image output by the token-based generator as:

$$G(\mathbf{z}) = g(f(\mathbf{z}), \mathbf{C}), \tag{4}$$

where $\mathbf{z} \in \mathcal{Z} \sim \mathcal{N}(0, 1)$ indicates the input latent code, $\mathbf{C} = \{\mathbf{c}_1, \cdots, \mathbf{c}_m\}$ denotes the sequence of learned constant content tokens through the input layer, and $f$, $g$, $G$ indicate the mapping network, the generator network and the token-based generator, respectively. Our token-based generator follows the same optimization objectives used by the style-based generator [27, 28], i.e., an non-saturating logistic adversarial loss with R1 regularization. To be specific, the generator loss is:

$$L_G = \mathbb{E}_{\mathbf{z} \sim \mathcal{N}(0,1)}[log(1 + exp(-D(G(\mathbf{z}))))], \tag{5}$$

where $D$ is the discriminator and the discriminator loss is

$$L_D = \mathbb{E}_{\mathbf{x} \sim P_d}[log(1 + exp(-D(\mathbf{x})))] + \mathbb{E}_{\mathbf{z} \sim \mathcal{N}(0,1)}[log(1 + exp(D(G(\mathbf{z}))))] + R_1(\psi), \tag{6}$$

where $x \sim P_d$ denotes images from the real data, and $R_1(\psi)$ is the regularization term calculated by:

$$R_1(\psi) = \frac{\gamma}{2} \mathbb{E}_{\mathbf{x} \sim P_d} \left[ \|\nabla D_\psi(x)\|^2 \right]. \tag{7}$$

Specifically, the $R_1$ regularization term [32] is computed less frequently than the main loss function following the commonly used lazy regularization strategy, thus greatly diminishing the computational cost and the overall memory usage [27, 28, 52].

# 4 Experiments

## 4.1 Experiment Setup

**Dataset** To evaluate the token-based generator and make fair comparisons with the SOTA approach (i.e., the style-based generator [28]), we conduct experiments on the most commonly-used public research datasets, i.e., Flickr-Faces-HQ (FFHQ) [27] and Large-scale Scene Understanding (LSUN) [47]. Specifically, FFHQ consists of 70,000 high-quality images of human face, which is able to evaluate the model's ability to synthesize high-frequency details. We conduct experiments on FFHQ with both $256 \times 256$ and $1024 \times 1024$ image size following previous works [15, 22]. Besides, we choose the church set of LSUN [47] to evaluate the synthesis in terms of complex scenes.

**Evaluation.** In this section, we conduct both quantitative and qualitative experiments on the token-based generator. Specifically, we report quantitative results by three commonly-used metrics, i.e., Fréchet Inception Distance (FID) [19], Precision and Recall [29, 39]. We use FID as it has been widely used in many generation tasks as an effective perceptual metric [4, 26, 27]. In addition to the image quality assessment, we use the metrics of Precision and Recall to evaluate the distribution learned by GAN. Specifically, the Precision metric intuitively measures the quality of samples from the generated images, while recall measures the proportion of real images that is covered by the learned distribution by GAN [39].

**Training details** We use 8 NVIDIA V100 GPUs for training each model. We build upon the public Pytorch implementation of StyleGAN2[2] [28]. For fair comparisons, we inherit all of the training details and parameter settings from the configuration E setup. All models are trained and report the best results under the same training iterations. For each iteration, we set the size of mini-batch as 32. We use the Adam solver [3] with the same momentum parameters $\beta_1 = 0$, $\beta_2 = 0.99$ to train both the generator and discriminator. We apply $R_1$ regularization for every 16 iterations.

---

[2]https://github.com/rosinality/stylegan2-pytorch

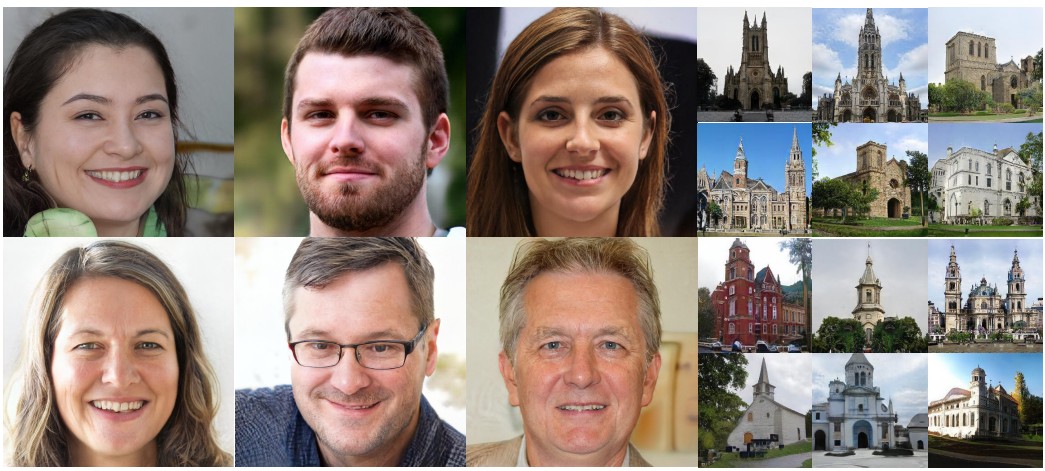

Figure 4: Uncurated results for FFHQ-1024 [27] and LSUN CHURCH [47]. The results show that token-based generator is able to synthesize both high-frequency details (e.g., hair and beard on the human face) and plausible structures for complex scenes (e.g., the church).

Table 1: Quantitative comparisons with the state-of-the-art model (i.e., StyleGAN2 [28]) on FFHQ-256 [27], FFHQ-1024[27] and LSUN CHURCH [47]. We compute each metric 10 times with different random seeds and report their average. The result shows that the token-based generator achieves comparable performance with the SOTA. ↑ the higher, the better. ↓ the lower, the better.

| Model | Metrics | FFHQ-256 [27] | FFHQ-1024 [27] | LSUN CHURCH [47] |
|---|---|---|---|---|
| StyleGAN2 [27] | FID↓ | $6.09 \pm 0.051$ | $5.20 \pm 0.049$ | $5.88 \pm 0.043$ |
| | Precesion↑ | $0.643 \pm 0.002$ | $0.653 \pm 0.002$ | $0.587 \pm .002$ |
| | Recall ↑ | $0.401 \pm 0.002$ | $0.411 \pm 0.002$ | $0.358 \pm 0.002$ |
| TokenGAN(Ours) | FID↓ | $5.41 \pm 0.050$ | $5.21 \pm 0.032$ | $5.56 \pm 0.037$ |
| | Precesion↑ | $0.660 \pm 0.002$ | $0.651 \pm 0.001$ | $0.577 \pm 0.002$ |
| | Recall↑ | $0.447 \pm 0.003$ | $0.442 \pm 0.004$ | $0.376 \pm 0.002$ |

### 4.2 Unconditional image synthesis

**Quantitative comparisons.** We report quantitative comparisons with StyleGAN2 [28] due to its state-of-the-art performance. For fair comparisons, we report the results with the same training iterations on FFHQ-256 [27], FFHQ-1024 [27] and LSUN CHURCH [47] by three objective metrics, i.e., FID, Precision and Recall in Table 1. The results show that the token-based generator has achieved the state-of-the-art results in terms of both perceived quality and distribution modeling.

**Qualitative results.** To demonstrate the image quality of the synthesis results for the token-based generator, we randomly sample noise $\mathbf{z} \in \mathcal{Z} \sim \mathcal{N}(0,1)$ to generate images for high-resolution images (i.e., with resolution $1024 \times 1024$) of human face and the images of complex church scenes. The uncurated results can be found in Figure 4. It shows that the token-based generator is able to synthesize both high-frequency details of the human face and plausible structures for complex scenes.

**Style editing.** We study and report the results of style editing by the TokenGAN in Figure 5. All the results are obtained by editing the style token of interest and then re-synthesizing images using edited style tokens. The result shows several interesting properties of the TokenGAN. First, the TokenGAN has a similar behavior as in StyleGAN2, i.e., controlling coarse/middle/fine styles at different layers (e.g., pose/hairstyle/color scheme in $8^2/16^2 - 64^2/128^2 - 256^2$ layers). Besides, the TokenGAN shows localized behaviors for the style tokens. For example, in the first row, editing the style token for the pose "globally" changes the pose while remaining the styles of other regions (e.g., hairstyle, background). In the second row, editing the style token for hair length, TokenGAN "locally" turns the hair longer while remaining other styles.

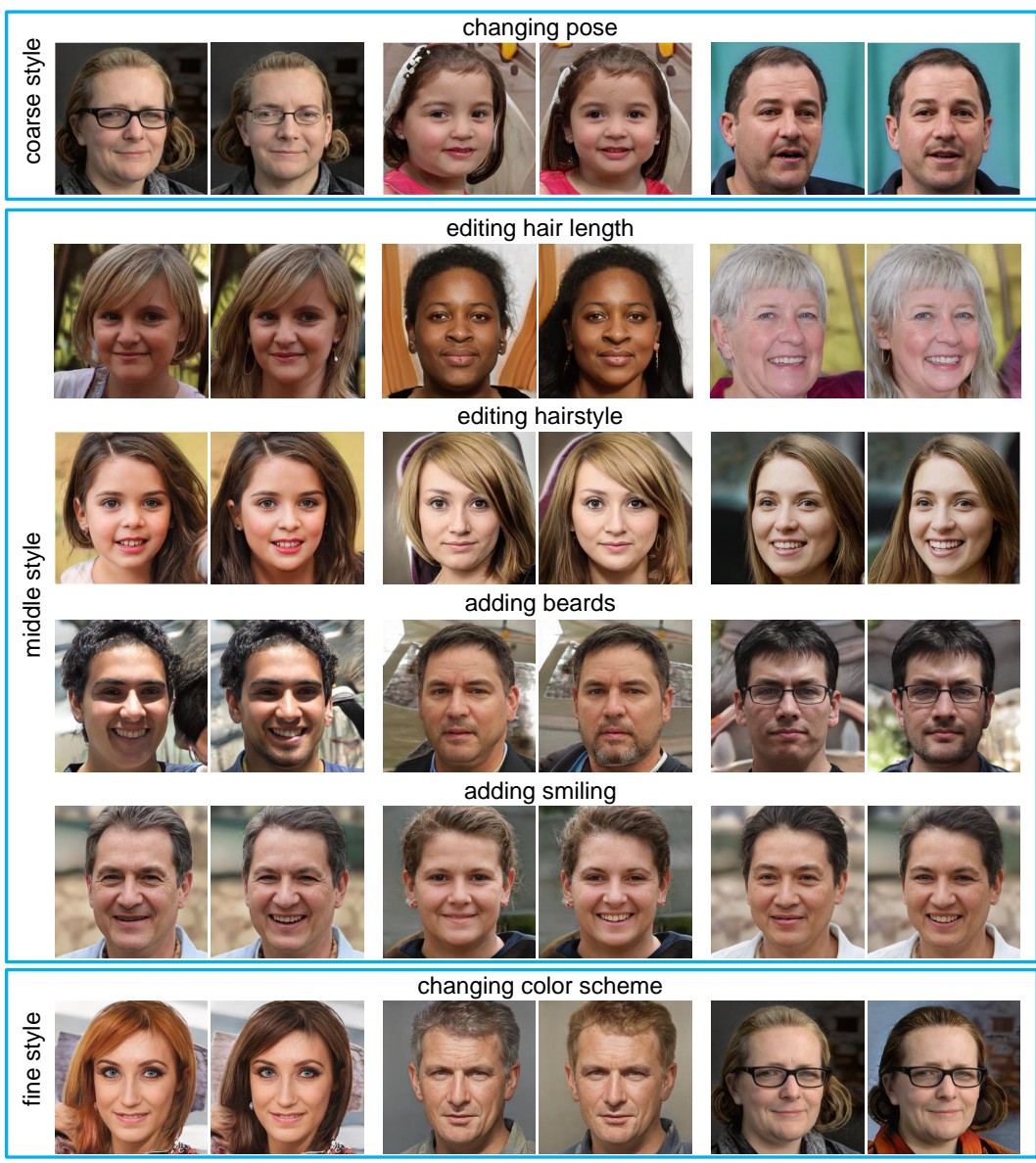

Figure 5: Results of style editing. All the results are obtained by editing the style token of interest and then re-synthesizing images using edited style tokens. It shows that TokenGAN controls coarse/middle/fine styles at different resolution layers (e.g., pose/hairstyle/color scheme in $8^2/16^2 - 64^2/128^2 - 256^2$ layers). Besides, the TokenGAN is able to get localized behavior by editing a specific style while remaining other styles at the same time. For example, in the second row, the TokenGAN turns the hair length longer without changing other styles.

**Image inversion.** To better apply well-trained GANs to real-world applications, the technique of GAN inversion has attracted an increasing attention [40, 53, 1, 2]. Such an inversion technique enables real image editing by searching the most accurate latent code in the learned latent space to recover the real image [53]. It has been demonstrated that the model with better inversion results tend to learn a better image modeling of the real data [28]. Specifically, we randomly sample several real images from FFHQ-256 [28] and adopt the optimization-based inversion technique used by most works [27, 40, 53]. For fair comparisons, we use the same setting following StyleGAN2 [27]. The inversion results by the style-based generator and the token-based generator are shown in Figure 6. It shows that the token-based generator is able to reconstruct real images better. For example, StyleGAN2 tends to reconstruct the lips and the headscarf in the fifth case with the same red color,

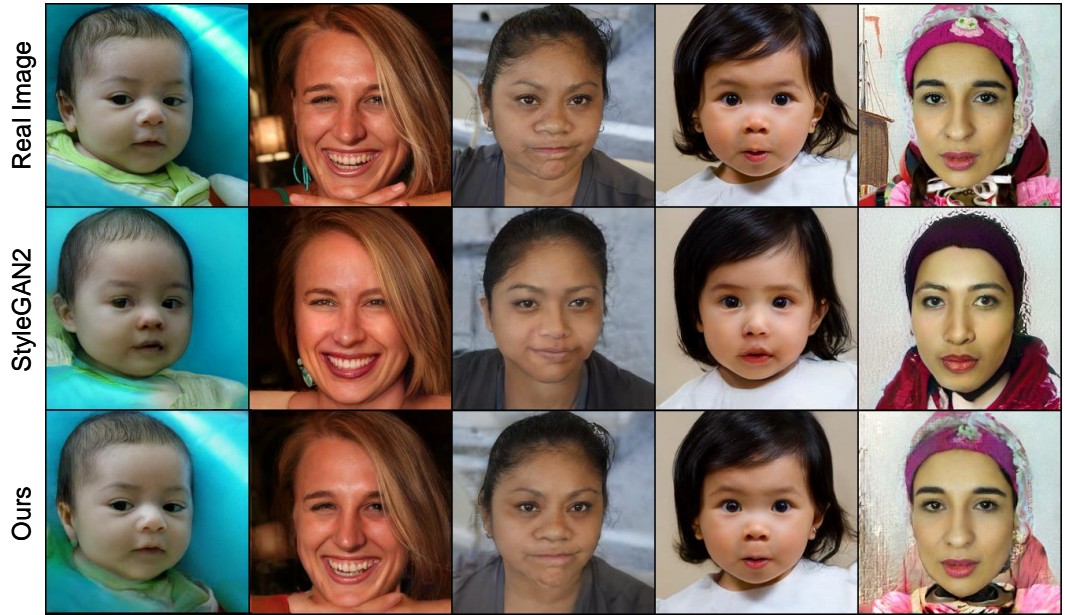

Figure 6: Visual results of image inversion by StyleGAN2 [28] and the token-based generator. We adopt the same inversion technique following the common paradigm [1, 2, 27]. It shows that the token-based generator is able to reconstruct fine-grained details with the dense style control over image regions (e.g., the facial expressions in the third case).

Table 2: Quantitative comparison of the reconstructed images by image inversion in terms of mean average error (MAE, range=[0,255]) and LPIPS distance [51].

| Model | MAE↓ | LPIPS ↓ | Model | MAE↓ | LPIPS ↓ |
|---|---|---|---|---|---|
| StyleGAN2[28] | 16.45 | 0.1539 | TokenGAN | **13.43** | **0.1238** |

while the token-based generator is able to reconstruct accurate colors for the lips and the headscarf respectively. This is because TokenGAN renders different regions by using different style tokens via attention mechanism, while StyleGAN2 renders them by a single style vector.

For a better quantitative comparison for image inversion, we randomly sample 1,000 real images from FFHQ-256 and report the mean absolute error (MAE, range=[0,255]) and the LPIPS distance [51] of the inversion results by StyleGAN2 and TokenGAN. The results in Table 2 show that TokenGAN is able to reconstruct significantly more accurate results with much lower MAE and shorter LPIPS distance, which is in line with the visual results in Figure 6.

**Image interpolation.** To further explore the property of the learned latent space by the token-based generator, we perform image interpolation by a linear interpolation in the learned intermediate latent space $\mathcal{S}$. In practice, we randomly sample two noises $\mathbf{z}_1, \mathbf{z}_2$ and run them through the mapping network to have their corresponding sequence of style tokens $s^1 := \{\mathbf{s}_1^1, \cdots, \mathbf{s}_n^1\}$, $s^2 := \{\mathbf{s}_1^2, \cdots, \mathbf{s}_n^2\}$. After that we perform linear interpolation for each token by $\mathbf{s}_i^3 = \alpha \times \mathbf{s}_i^1 + (1 - \alpha) \times \mathbf{s}_i^2, \alpha \in (0, 1)$ and use the new style tokens $s^3 := \{\mathbf{s}_1^3, \cdots, \mathbf{s}_n^3\}$ to synthesize a new image. As shown in Figure 7, we show the sampled styles $s^1, s^2$ in the first column and the last column, and the interpolated styles between them. The results show that the token-based generator is able to perform perceptually smooth transition between two different styles (e.g., glasses, age, hair length, coloring, etc.)

### 4.3 Ablation study

To study the effectiveness of each component of the token-based generator, this section presents ablation studies on FFHQ-256 [27] after 0.4 million training iterations in terms of FID [19]. Specifically, we calculate the metric 10 times with different random seed and report the average results.

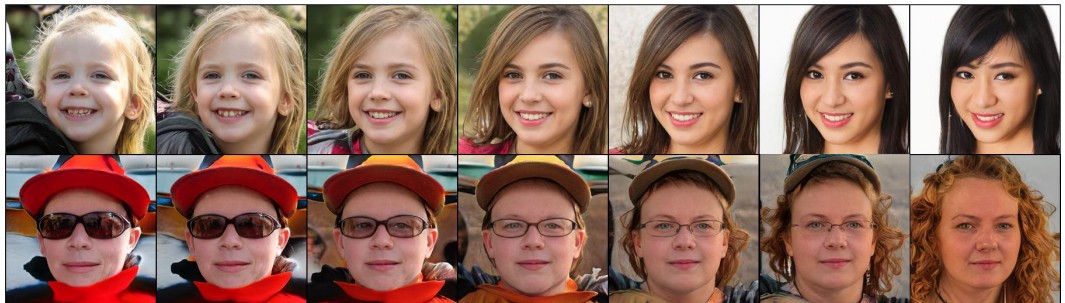

Figure 7: Results of image interpolation by the token-based generator. In practice, we sample two latent codes (the 1st and the 7th column) and perform linear interpolation of them in the learned intermediate latent space (interpolated results are from the 2th to the 6th column). The token-based generator shows smooth transition of high-level attributes in results (e.g., glasses in the second row).

Table 3: Quantitative ablation study on the number of style tokens.

| style tokens | 8 | 16 | 32 | 64 |
|---|---|---|---|---|
| FID | $7.66 \pm 0.060$ | $7.02 \pm 0.060$ | $\mathbf{6.81 \pm 0.044}$ | $7.60 \pm 0.071$ |

**The number of style tokens.** To study the effectiveness of the style tokens, we conduct ablation studies by using different number of style tokens. As shown in Table 3, with the growing number of the style tokens, there are more style tokens to model the distribution of styles for different semantics and the quality of images are improved. However, when the number of style tokens grows to 64, the learning will be difficult and performance would drop.

Table 4: Comparisons on the content tokens.

| $(m_{64}, m_{128}, m_{256})$ | FID$\downarrow$ |
|---|---|
| $(16^2, 16^2, 32^2)$ | $18.54 \pm 0.087$ |
| $(32^2, 32^2, 64^2)$ | $15.69 \pm 0.067$ |
| $(64^2, 64^2, 128^2)$ | $\mathbf{6.81 \pm 0.044}$ |

Table 5: Comparisons on the style normalization.

| Model | FID |
|---|---|
| InstanceNorm [41] | $13.7 \pm 0.051$ |
| PixelNorm [25] | $6.96 \pm 0.050$ |
| LayerNorm [43] | $\mathbf{6.81 \pm 0.044}$ |

**The number of content tokens.** We empirically set the number of content tokens $m$ at different layers according to the image size for the best performance and training efficiency. We conduct ablation study from layer $64^2$ to $256^2$ and denote the number of content tokens in these layers as $(m_{64}, m_{128}, m_{256})$. The results in Table 4 show that more content tokens could provide more fine-grained control over the whole images, leading to better results in image generation.

**Style normalization layers.** We compare different style normalization layers in Table 5. The results show that, PixelNorm [25] and LayerNorm [43] significantly outperforms the style normalizatioin by instance normalization in the token-based generator [41]. We choose LayerNorm as the style normalization layer in the token-based generator.

## 5  Conclusion

In this paper, we propose a novel TokenGAN showing that the token-based representation of image features and styles could enable content-aware and fine-grained style learning for image synthesis. Specifically, we use Transformer to model the interaction between content tokens and style tokens, which facilitates the perceived quality of generated images and shows promising properties in terms of style editing. In the future, we will study to extend the generative transformers to extensive applications, e.g., style transfer [20], animation generation [45], image inpainting [49], etc.

## Acknowledgments and Disclosure of Funding

Funding in direct support of this work: NSF of China under Grant 61672548 and U1611461, GPUs provided by Microsoft Research Asia. Additional revenues related to this work: internship at Microsoft Research Asia.

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
