# OpenReview forum: " Improving Visual Quality of Image Synthesis by A Token-based Generator with Transformers"
_NeurIPS.cc/2021/Conference — NeurIPS 2021 Poster_

### Official Review · Reviewer_whGT · 2021-07-12

**Rating:** 6
**Confidence:** 4

**Summary:**

This paper investigates the application of transformer for image generation by replacing the convolutional layer with MLP(including attention). The motivation provided is the limitation of the style-based generator (e.g.,  StyleGAN), which leverages the single style over multiple image regions, allowing a less entangled representation learning in W and visible artifacts in generated images.  To combat this shortcoming, authors introduce transformer-liked architecture, and successfully generate high-quality images (1024*1024) without touching any convolutional layers in generator.

**Ethics Review Area:**

["I don’t know"]

**Limitations And Societal Impact:**

Cons:
The proposed method maybe not corresponding to the provided motivation(the limitation of the stylegan), since the proposed method is based on the styleganv2, and leverages all the techniques of StyleGANv2. I am wondering the motivation of the proposed method. In fact, I guess the motivation is to use firstly transformer to generate high-fidelity image.

I am hoping that more analysis or insights could be exhibited in this paper, since the transformer is firstly used in generation model. However, all shown and reported results are little boring, although it looks better than styleganv2. In fact, this paper is based styleganv2.

The transformer is only exploited in generator instead of discriminator. The related work has already shown that the transformer obtains outstanding performance in classification task, why in this paper authors do not use it in discriminator.

Since the learnable position encoding is intruded in content tokens, I am wondering why the position encoding do not be injected in style token? Besides, I fail to find the code presenting the position encoding when checking the provided code. Please correct me if I am wrong.

The detail for the value of m and k are loss, even in supplementary material.  Is the model performance influenced by these selected values?


**Main Review:**

Pros:
Authors firstly exploit the image generation with transformer to synthesize high-fidelity images. The transformer-liked architecture has been used a series of computer vision tasks,  in this paper authors attempt to introduce it into image generation tasks. To achieve better performance, authors devise the model based on the StyleGANv2, aiming to use the well-devised framework and techniques.

One of the interesting parts is to generate the high-quality image without using any convolutional layers, which exhibits the power of the transferor-liked architecture.  Comparing StyleGANv2, the proposed method is able to obtain better performance in some metrics.

The proposed method can be further used for other tasks (image editing), which almost keep the property of the StyleGAN.

**Time Spent Reviewing:**

3.5

---

> ### Author Response · Authors · 2021-08-07
> **Motivation, Transformer, position encoding and implementation details.**
>
>
> We sincerely appreciate the reviewer’s time, effort, and expertise. We address major concerns as below.
>
> **1. Motivation.** We emphasize our motivation as introducing a new perspective of image synthesis by using **a token-based generator**. Under this perspective, we naturally introduce the content tokens, the style tokens, and the content-aware style modulation method based on Transformer. Compared with StyleGAN, such a content-aware style modulation enables a more fine-grained style controllability of images.
>
> As shown in [**[the link to the results of StyleGAN]**](https://drive.google.com/file/d/1U-nlMDtpnf1RcYkaFQtbh5oxnhA97hy6/view?usp=sharing), in the second row, StyleGAN changes the pose of Source A to Source B by using coarse styles from B but also changes the gender of A. In contrast, as shown in [**[this link to the results of TokenGAN on FFHQ256]**](https://www.dropbox.com/s/y1jkkqjkuc6tr19/style_editing_v0.png?dl=0), TokenGAN is able to change the pose by editing the style token for the pose in the first row while remaining other styles unchanged in our results. All the results are obtained by editing the style token of interest and then re-synthesizing images using edited style tokens.
>
>
>
> **2. Transformer-based discriminator.** In this paper, our focus is the **design of the generator** and we follow previous works [26, 17] to use a convolution-based discriminator for training. We're willing to explore the Transformer-based discriminator in our future work.
>
> **3. Position encoding.** Thank you for pointing out this issue. We have mistakely submitted an old version of codes. We will correct the codes in our final version. The codes of the final model for the position encoding is shown as below.
>
> ```
> class ModulatedConv(nn.Module):
>     def __init__(self, in_size, style_dim, style_mod='prod', norm_type='layernorm'):
>         super().__init__()
>
>         ires, in_channel, self.ipsize, style_num = in_size
>         self.style_mod = style_mod
>         self.norm_type = norm_type
>
>         self.keys = nn.Parameter(
>             nn.init.orthogonal_(torch.empty(1, style_num, in_channel)))
>         self.pos = nn.Parameter(torch.zeros(1, (ires // self.ipsize) ** 2, in_channel))
>         self.attention = MultiheadAttention(in_channel, in_channel, in_channel, style_dim)
>
>     def forward(self, input, style):
>         b, t, c = input.size()
>
>         # remove old style
>         input = norm(input, norm_type=self.norm_type)
>         input = input.view(b, t, -1, self.ipsize, self.ipsize)
>
>         # calculate new style
>         # multi-head attention
>         query = torch.mean(input, dim=[3,4])
>         keys = self.keys.repeat(input.size(0), 1, 1)
>         pos = self.pos.repeat(input.size(0), 1, 1)
>         new_style, _ = self.attention(q=query+pos, k=keys, v=style)
>
>         # append new style
>         if self.style_mod == 'prod':
>             out = input * new_style.unsqueeze(-1).unsqueeze(-1)
>         elif self.style_mod == 'plus':
>             out = input + new_style.unsqueeze(-1).unsqueeze(-1)
>         else:
>             raise NotImplementedError(
>                 'Have not implemented this type of style modulation')
>
>         out = out.view(b, t, c)
>         return out
> ```
>
> Specifically, we use position encoding for content tokens to supplement the position information from 2D images following previous works [6,7,8,13], while the style tokens are position-invarinat, which do not need position encoding.
>
> **4. Implementation details.** We will incorporate the following details in the final version. The number of content tokens $m$ is growing progressively with layers for different resolutions (from $8^2$ to $1024^2$). We empirically set the dimensions of the content tokens $(p, m, d)$ at different layers according to the image size for the best performance and training efficiency. The details are listed below, which are also specified in our submitted codes.
>
> | spatial size | p | m | d |
> | :-: | :-: | :-: | :-: |
> | $8^2$   | 1 | $(8/1)^2$  | 512
> | $16^2$  | 1 | $(16/1)^2$ | 512
> | $32^2$  | 1 | $(32/1)^2$ | 512
> | $64^2$  | 1 | $(64/1)^2$ | 256
> | $128^2$ | 2 | $(128/2)^2$ |128
> | $256^2$ | 2 | $(256/2)^2$ |64
> | $512^2$ | 4 | $(512/4)^2$ |32
> | $1024^2$ |4 | $(1024/4)^2$ |16
>
> We conduct ablation study on FFHQ-256 dataset from Layer $64^2$ to $256^2$ and denote the number of content tokens in these layers as $(m_{64}, m_{128}, m_{256})$. The results in the following table show that more content tokens could provide more fine-grained control over the whole images, leading to better results in image generation.
>
> | $(m_{64}, m_{128}, m_{256})$ | FID |
> | :-: | :-: |
> |$(16^2, 16^2, 32^2)$  | 18.54 $\pm$  0.087|
> |$(32^2, 32^2, 64^2)$  | 15.69 $\pm$  0.067 |
> |$(64^2, 64^2, 128^2)$ | **6.81 $\pm$ 0.044** |

---

> > ### Comment · Reviewer_whGT · 2021-08-27
> > **Concerns**
> >
> > Thanks authors for rebuttal. I have some concerns:(1) The motivation. As descripted in 24-35 lines and 36 lines, the motivation is to address the shortcoming of Stylegan.  However, authors' feedback is not corresponding to this context presented in main paper.  Please explain information of 24-36 lines.  (2) The discriminator. I would like to believe that authors explored s series of transformer-based discriminator, since it is weird the discriminator is convolutional network and the generator is transformer. I guess the result maybe be not interesting when devising transformer-based GAN.

---

> > > ### Author Response · Authors · 2021-08-29
> > > **The motivation and the discriminator**
> > >
> > >
> > > We appreciate your prompt feedback, and we address the remaining concerns below.
> > >
> > > **1. Motivation.**
> > >
> > > 1) In the paper, we chose to introduce our motivation from the aspect of exiting issues. Since StyleGAN controls the style by imposing a single style code to all the regions of a feature map within the same layer, neglecting that different regions may have different semantics that needs learning different style codes (as claimed in Line 31-34). Typically, better image modeling requires learning different style codes to control hairstyle and beard style separately. This issue motivates us to design a content-aware and fine-grained style control for different regions (as claimed in Line 36-37). As illustrated in Figure 1-(c) in the paper, we introduce the new perspective by a token-based generator (TokenGAN). The TokenGAN decomposes the feature map **c** into a sequence of content tokens and uses a sequence of style tokens **s** to control them. In practice, we choose the cross-attention mechanism used in Transformer to establish the content-aware style control due to its powerful sequence modeling (Line 50-52).
> > >
> > > 2) In the rebuttal, we want to clarify that our motivation is **NOT simply to firstly use the popular Transformer for image generation**. We emphasize our motivation as introducing the new perspective of using a token-based generator to establish the content-aware style control to get rid of the issue of StyleGAN's style control. We believe this motivation is in line with the motivation claimed in the paper.
> > >
> > > 3) We have conducted extensive experiments to verify our motivation. First, we have shown the quantitative improvements of image synthesis in Table 2. Second, we have shown significant qualitative improvements of image inversion in Figure 5. In addition, we report the mean absolute error (MAE, range=[0,255]) and the LPIPS distance of the inversion results from StyleGAN2 and TokenGAN. Specifically, we randomly sample 1,000 real images from FFHQ-256 and conduct image inversion using the same projection method and hyperparameters following StyleGAN2 [27] (P8 Line 237-239). The results in the table below show that TokenGAN is able to reconstruct significantly more accurate results with much lower MAE and shorter LPIPS distance. We will add the quantitative results in the final version.
> > >
> > >    | Model | MAE$\downarrow$ | LPIPS$\downarrow$ |
> > >    | :-: | :-: | :-:|
> > >    |StyleGAN2 | 16.45 | 0.1539 |
> > >    |TokenGAN  |**13.43**|**0.1238**|
> > >
> > >     Finally, we have compared the style editing results of TokenGAN ([link](https://www.dropbox.com/s/y1jkkqjkuc6tr19/style_editing_v0.png?dl=0)) and StyleGAN ([link](https://drive.google.com/file/d/1U-nlMDtpnf1RcYkaFQtbh5oxnhA97hy6/view?usp=sharing)). As we introduced in the rebuttal, TokenGAN has shown much finer style controllability over StyleGAN. For example, in the second row from the results of StyleGAN, StyleGAN changes the pose of the woman from the side to the front but also changes the gender to male. In contrast, TokenGAN is able to change the pose of the man in the first row by editing only the style token for the pose while remaining other styles unchanged.
> > >
> > > Upon reading your comments, we will revise it and make it clear in the final version.
> > >
> > > **2. Transformer-based discriminator.**
> > >
> > > Since our motivation is introducing a token-based generator (i.e., TokenGAN) to get rid of the issue of StyleGAN's style control, our focus is the design of the generator, and we simply follow the previous design ([26, 27]) for the discriminator. It is not uncommon that adopting different designs for the generator and the discriminator. For example, StyleGAN introduces a mapping network in the generator but they didn't use it in the discriminator. StyleGAN2 introduces weight normalization in the generator but they use vanilla convolutions in the discriminator.

---

### Official Review · Reviewer_SrfN · 2021-07-15

**Rating:** 5
**Confidence:** 3

**Summary:**

In this paper, a transformer-based modification of the generator in the ubiquitous StyleGAN model is proposed (named _TokenGAN_). More specifically, the entire generator is replaced by a convolution-free architecture where style-content modulation is now modeled using cross-attention layers instead of AdaIN. Similar to the original StyleGAN, this should allow for decoupled control of style and content at different granularities. The paper shows that the proposed approach achieves FID scores comparable to or better than StyleGAN2 for unconditional image synthesis on FFHQ and LSUN-Churches.

**Limitations And Societal Impact:**

Not discussed in the paper.

**Main Review:**

__Strengths:__

The proposed approach is a simple, yet working modification of the generator of StyleGAN. The use of cross-attention to model style-content interactions is a good, novel idea. The attention maps depicted in Figure 3 nicely demonstrate that the generator trained on FFHQ attends to human-interpretable concepts. The paper thoroughly reports error bars / standard deviations where applicable.

__Weaknesses:__

Although the paper states the "fine-grained controllability of visual tokens" (l.60), no experiments are made to explicitly verify this claim or how controllability is increased over the StyleGAN baseline.
Furthermore, the abstract states that the approach is "able to synthesize high-fidelity images [...], dispensing with convolutions entirely" (l.15-16). However, as the discriminator is the same as for StyleGAN2, this statement does only hold for the architecture of the generator, but not the complete model. Following on from this, it would be interesting to see if it is also possible to replace the discriminator with a pure Transformer architecture.

Given that the proposed architecture is mostly transformer-based, are there any notable differences in applicability to other datasets? For example, how does the model handle (very) small datasets or large-scale, complex datasets? Is overfitting a problem? If so, how is this being adressed? Here, plotting nearest neighbors (e.g. in VGG space) would be an interesting analysis option.

Finally, the paper does not mention on which patch-size $p$ the models are trained. How do different $p$ affect the results? Furthermore, how do different $m$ affect the model? I can only find a study regarding $n$. Also: Is $\mathbf{K}$ shared across different style modulation layers?

__Clarity:__
The presentation of the cross-attention layer and style/content tokens is clear; however, the "upsampling procedure" (l. 20-21 in the supplementary) remains unclear to me and should probably be included in a Figure (e.g. Fig. 2)

__Significance/Originality:__
Replacing the convolutions in the generator of StyleGAN has the potential to allow applications to more complex datasets than the often used FFHQ or single-scene LSUN variants. Unfortunately, this remains unexplored in this work, but might be of interest to the community.

__Justification of Rating:__
I think the use of cross-attention as a modulation level for style content is an interesting idea and is sufficiently justified in the paper. However, apart from the visualization of attention weights in Figure 3, the paper does not show how the proposed method differs in applicability compared to StyleGAN(2) and what the impact is of switching to a transfomer architecture. For example, how do the suggested modifications alter convergence speed? What's the effect when trained on smaller/larger datasets? Is overfitting a larger/smaller problem than in the underlying work? Including the weaknesses mentioned above, I currently think that the paper (in its current state) would be a good fit for a workshop, but needs to be expanded (see the comments above) to be published as a full conference paper. However, I am curious about the authors' response and the opinions of the other reviewers, and I am happy to be proven wrong.

_Minor_:
- l. 53: i -> in
- l. 63: decompose -> decomposes
- l. 69: missing space after "TokenGAN"
- l. 135: missing space after "TokenGAN"
- l. 186: adopts -> adopted
- l. 268: view -> viewing
- l. 278: demonstrate -> demonstrated
- Fig. 2: "concate" -> "concatenate"/"concat"



__[UPDATE] Post-Rebuttal__

After re-reading the paper and rebuttal, I think my original rating was too harsh, and I am therefore raising my score to 5. However, I am still not convinced that the proposed approach offers many advantages over the convolution-based StyleGAN(2) architecture, i.e., the gains presented here are rather marginal.  I think the work would benefit from a more fundamental analysis of the differences (training dynamics, applicability) between TokenGAN and StyleGAN, where then an improvement in the typical metrics (e.g. FID) would be nice but not mandatory.

**Time Spent Reviewing:**

7.5

---

> ### Author Response · Authors · 2021-08-07
> **Comparisons with StyleGAN, discussions on Transformer architecture and implementation details**
>
>
> We sincerely appreciate the reviewer for the acknowledgment of the TokenGAN as a "simple yet working" approach, the use of cross-attention is "good and novel" and the idea is "sufficiently justified in the paper". We thank the reviewer’s time, effort, and expertise. We address major concerns as below.
>
> **1. Comparisons with StyleGAN.** As shown in [**[the link to the results of StyleGAN]**](https://drive.google.com/file/d/1U-nlMDtpnf1RcYkaFQtbh5oxnhA97hy6/view?usp=sharing), in the second row, StyleGAN changes the pose of Source A to Source B by using coarse styles from B but also changes the gender of A. In contrast, in [**[this link to the results of TokenGAN on FFHQ]**](https://www.dropbox.com/s/y1jkkqjkuc6tr19/style_editing_v0.png?dl=0), TokenGAN is able to change the pose by editing the style token for the pose in the first row while remaining other styles unchanged. The results have demonstrated that TokenGAN has finer-grained style controllability than StyleGAN. All the results of TokenGAN are obtained by editing the style token of interest and then re-synthesizing images using edited style tokens. We will incorporate the results in the final version.
>
> We emphasize TokenGAN as **a token-based generator** (P1 Line 7), which is convolution-free. In this paper, our focus is the **design of the generator** and we follow previous works [26, 17] to use a convolution-based discriminator for training. We're willing to explore the Transformer-based discriminator in our future work.
>
>
> **2. Discussion on Transformer.**  We will incorporate the following clarifications in the final version.
>
> 1) Overfitting. Overfitting is not a problem in TokenGAN. As proven in previous works, pure GAN models appear to generalize well and overfitting was undetectable even with the state-of-the-art detectors (Webster, Ryan, et al. Detecting overfitting of deep generative networks via latent recovery. In CVPR 2019). Besides, as shown in the Precision results in Table 2, existing GAN models are still far from overfitting to the training data (which should have an extremely high precision value).
>
> 2) Training details. By switching to a transformer-based architecture, it takes about 52 hours to train TokenGAN with eight 16G NVIDIA V100 on FFHQ-256, in comparison with StyleGAN2's 46 hours. Besides, we found the dataset-specific regularization setting is very helpful in training. Specifically, when training on a smaller dataset (e.g., FFHQ-256 [26]), we use a smaller weight (e.g., r1=1) of R1 regularization (P7, Line 221). When training on a larger and more complex dataset (e.g. LSUN dataset [44]), the learning of the generator becomes more challenging thus we use large R1 regularization (e.e., r=100) to regularize the discriminator. We report the inference time of generating an image by TokenGAN and StyleGAN2 in the table below. Specifically, we test on a single NVIDIA V100 GPU 100 times and report their average and error bars.
>
> | Model | resolution | inference time (ms) |
> | :-: | :-: | :-: |
> | StyleGAN2   | $1024^2$ | 22.33 $\pm$ 0.17
> | TokenGAN    | $1024^2$ | 26.35 $\pm$ 0.09
> | StyleGAN2   | $256^2$ | 15.83 $\pm$  0.04
> | TokenGAN    | $256^2$ | 16.10 $\pm$  0.07
>
>
> **3. Implementation details**. We will incorporate the following details in the final version.
>
> 1) Upsampling procedure. We follow StyleGAN2 and adopt the skip generator architecture in TokenGAN. Such a skip generator generates images in each layer at different resolutions (e.g., from $4^2$ to $1024^2$) and progressively upsamples and sums the images from the previous layer by skip connections to the next layer. After progressive summing, the generator takes the output in the last layer as the final results. Specifically, in each layer of TokenGAN, we pack the sequence of style-modulated content tokens together and reshape them to a 2D shape as the images.  With $1024^2$ output resolution, the token-based generator contains a total of 15 layers of style blocks, where the first one corresponds to $8^2$ resolution, the next two correspond to $16^2$, and so forth. Such a design enables TokenGAN inherit the property that generates coarse style in low-resolution layers and generates fine-grained details in high-resolution images.
>
> 2) Since different style modulation layers focus on different scales of styles (e.g., coarse style in low-resolution layers and fine details in high-resolution layers), the learnable semantic embedding $K$ is not shared in different layers.
>
> 3) The dimension of content tokens. The number of content tokens $m$ is growing progressively with layers for different resolutions (from $8^2$ to $1024^2$). We empirically set the dimensions of the content tokens $(p, m, d)$ at different layers according to the image size for the best performance and training efficiency. The details are listed below, which are also specified in our submitted codes.
>
> | spatial size | p | m | d |
> | :-: | :-: | :-: | :-: |
> | $8^2$   | 1 | $(8/1)^2$  | 512
> | $16^2$  | 1 | $(16/1)^2$ | 512
> | $32^2$  | 1 | $(32/1)^2$ | 512
> | $64^2$  | 1 | $(64/1)^2$ | 256
> | $128^2$ | 2 | $(128/2)^2$ |128
> | $256^2$ | 2 | $(256/2)^2$ |64
> | $512^2$ | 4 | $(512/4)^2$ |32
> | $1024^2$ |4 | $(1024/4)^2$ |16
>
> We conduct ablation study on FFHQ-256 dataset from Layer $64^2$ to $256^2$ and denote the number of content tokens in these layers as $(m_{64}, m_{128}, m_{256})$. The results in the following table show that more content tokens could provide more fine-grained control over the whole images, leading to better results in image generation.
>
> | $(m_{64}, m_{128}, m_{256})$ | FID |
> | :-: | :-: |
> |$(16^2, 16^2, 32^2)$  | 18.54 $\pm$  0.087|
> |$(32^2, 32^2, 64^2)$  | 15.69 $\pm$  0.067 |
> |$(64^2, 64^2, 128^2)$ | **6.81 $\pm$ 0.044** |

---

> > ### Comment · Reviewer_SrfN · 2021-08-25
> > **Thanks!**
> >
> > Hello,
> >
> > Thank you very much for the clarifications and revisions. I have added an update to my review.

---

> > > ### Author Response · Authors · 2021-08-29
> > > **Advantages over the CNN-based StyleGAN(2) architecture.**
> > >
> > > We appreciate your prompt response and feedback, and we would like to address the remaining concern about the advantages over CNN-based StyleGAN2 as below.
> > >
> > > **1. Significant better image inversion.**  In addition to the improvements in terms of FID verified in Table 2, we follow styleGAN2 [27] and adopt the image inversion technique to measure the generative capability of different models. The better the reconstruction results, the stronger the model (as demonstrated in [27, 28, 34]). As shown in Figure 5 in the paper, TokenGAN shows significantly better inversion results compared with StyleGAN2.
> > > For example, in the fifth case, TokenGAN is able to render the clothe and the lip with different pinks that are closer to the input image while StyleGAN tends to render them with the same pinks. This is because TokenGAN renders different regions by using different style tokens via attention mechanism, while StyleGAN2 renders them by a single style vector.
> > >
> > > For a better quantitative comparison for image inversion, we report the mean absolute error (MAE, range=[0,255]) and the LPIPS distance of the inversion results from StyleGAN2 and TokenGAN. In practice, we randomly sample 1,000 real images from FFHQ-256 and conduct image inversion using the same projection method and hyperparameters following StyleGAN2 [27] (P8 Line 237-239). The results in the table below show that TokenGAN is able to reconstruct significantly more accurate results with much lower MAE and shorter LPIPS distance, which is in line with the visual results in Figure 5. We will add the quantitative results in the final version.
> > >
> > > | Model | MAE$\downarrow$ | LPIPS$\downarrow$ |
> > > | :-: | :-: | :-:|
> > > |StyleGAN2 | 16.45 | 0.1539 |
> > > |TokenGAN  |**13.43**|**0.1238**|
> > >
> > >
> > > **2. Finer style controllability by TokenGAN ([link](https://www.dropbox.com/s/y1jkkqjkuc6tr19/style_editing_v0.png?dl=0)) over StyleGAN ([link](https://drive.google.com/file/d/1U-nlMDtpnf1RcYkaFQtbh5oxnhA97hy6/view?usp=sharing)).**
> > >
> > > We would like to highlight that TokenGAN has shown much finer style controllability over StyleGAN.
> > > Specifically, StyleGAN enforces a single style vector to control all the regions of a feature map within the same layer, neglecting that regions are with different semantics, and thus leading to style entanglement for different regions (e.g., pose and gender in low-resolution layers). For example, in the second row from the results of StyleGAN, StyleGAN changes the pose of the woman from the side to the front but also changes the gender to male.
> > >
> > > In contrast, TokenGAN is able to change the pose by editing only the style token for the pose while remaining other styles unchanged. For example, in the first row from the results of TokenGAN, it is able to change the pose of the man from the side to the front while remaining the gender unchanged.
> > >
> > > In this paper, we present a new perspective by introducing such a token-based generator, which shows significantly better inversion property and finer style controllability, which are two important properties for using off-the-shelf GANs for image editing [1,2,38,48]. We believe it would make a valuable contribution and generate an interesting and important discussion within the community.
> > >
> > >
> > > Best regards.

---

### Official Review · Reviewer_cPNP · 2021-07-16

**Rating:** 6
**Confidence:** 4

**Summary:**

This paper proposes TokenGAN for unconditional image synthesis. The token-based generator takes the learned content tokens and style tokens from the latent space as input, and output a series of style-modulated content tokens, which are then concatenated and reshaped to get the output image. Cross-attention is used to assign the styles to different content tokens. The model adopts the style-based generator idea from StyleGAN and introduces the recent token-based structures and transformers to the traditional style-based generator. Experiments are conducted on FFHQ and LSUN CHURCH datasets and the quantitative results are slightly better than StyleGAN2.

**Ethical Concerns:**

There are no ethical concerns.

**Limitations And Societal Impact:**

The authors do not address the limitations and potential negative societal impact of their work. It would be good to add some discussions about  the potential limitations and failure cases of this work.

**Main Review:**

Strengths:
1. The proposed approach is well-motivated. It takes advantage of the latent code mapping and style-based modulation of StyleGAN, and further improves it by replacing the convolution-based architecture with token-based cross-attention.
2. By using a series of content tokens and style tokens, each patch on the image can adaptively attend to content-aware styles. So the structure can control the styles of different patches, in contrast to a single global style code in StyleGAN.

Weakness:
1. The quantitative results (FID, precision, and recall) are only slightly better than StyleGAN2. The qualitative results only show the generated images of the proposed TokenGAN, but does not show the comparison with StyleGAN2. From the generated images shown on the paper, I did not observe very significant improvements from StyleGAN2.
2. Some ablation studies are missing: (1) the number of content token (2) Mixing different style tokens (used in this paper) v.s. mixing styles at different layers (used in StyleGAN) for style mixing. (3) Style modulation by amplifying each channel (in this paper) v.s. by adaptive normalization (in StyleGAN).
3. For the discussion on image inversion, evaluation metrics such as LPIPS distance should be reported and compared with StyleGAN2.
4. I am curious about the complexity and efficiency of the proposed model compared with StyleGAN2.
5. It would be good to visualize how the style tokens can impact different regions (impact tokens) of the images. The authors tried to visualize the attention maps in Fig. 3, but I also expect other visualizations or evaluations illustrating how the multiple style tokens and cross-attention benefit the generated image quality and attribute disentanglement. For example, a visualization of style mixing might indicate how changing one style token (and keeping other style tokens unchanged) can affect the synthesized images.
6. Lines 174 - 178 on implementation details are not clear to me. Could the authors explain more on how to incorporate the skip connections and how to do upsampling in the generator? Are there interactions (such as self-attention) between the content tokens? If not, how does the generator deal with global consistency and context information?
7. This paper lacks discussion and comparison with a similar work [21] which also uses transformer for GANs.

**Time Spent Reviewing:**

3 hours

---

> ### Author Response · Authors · 2021-08-07
> **Experimental and implementation details**
>
>
> We sincerely appreciate the reviewer for the acknowledgment of the design of content-aware style control and considering TokenGAN as a "well-motivated" approach. We thank the reviewer’s time, effort, and expertise. We address major concerns as below.
>
> **1&3. Comparisons with StyleGAN2.** In addition to the results in Table 2 and Figure 4, we follow styleGAN2 [27] and adopt the image inversion technique to measure the generative capability of different models. The better the reconstruction results, the stronger the model (as demonstrated in [27, 28, 34]). As shown in Figure 5, TokenGAN shows significantly better inversion results compared with StyleGAN2 (e.g., more accurate colors for the fifth case), which demonstrates the stronger capacity of TokenGAN. Per your suggestions, we report the mean absolute error (MAE, range=[0,255]) and the LPIPS distance of the inversion results from StyleGAN2 and TokenGAN as below.
>
> | Model | MAE$\downarrow$ | LPIPS$\downarrow$ |
> | :-: | :-: | :-:|
> |StyleGAN2 | 16.45 | 0.1539 |
> |TokenGAN  |**13.43**|**0.1238**|
>
> Specifically, we randomly sample 1,000 real images from FFHQ-256 and conduct image inversion using the same projection method and hyperparameters following StyleGAN2 [27] (P8 Line 237-239). The results show that TokenGAN is able to reconstruct more accurate results with much lower MAE and shorter LPIPS distance, which is in line with the visual results in Figure 5. Per your suggestions, we will include the results in the final version.
>
>
> **2. Ablation results.** We will incorporate the following ablation results in the final version per your suggestions.
>
> 1) **The number of content tokens $m$** is growing progressively with layers for different resolutions (from $8^2$ to $1024^2$). Specifically, with $1024^2$ output resolution, the token-based generator contains a total of 15 layers of style blocks, where the first one corresponds to $8^2$ resolution, the next two correspond to $16^2$, and so forth.
> We empirically set the dimensions of the content tokens  $(p, m, d)$ in different layers according to the image size for the best performance and training efficiency. The details are listed below, which are also specified in our submitted codes.
>
> | spatial size | p | m | d |
> | :-: | :-: | :-: | :-: |
> | $8^2$   | 1 | $(8/1)^2$  | 512
> | $16^2$  | 1 | $(16/1)^2$ | 512
> | $32^2$  | 1 | $(32/1)^2$ | 512
> | $64^2$  | 1 | $(64/1)^2$ | 256
> | $128^2$ | 2 | $(128/2)^2$ |128
> | $256^2$ | 2 | $(256/2)^2$ |64
> | $512^2$ | 4 | $(512/4)^2$ |32
> | $1024^2$ |4 | $(1024/4)^2$ |16
>
> We conduct ablation study on FFHQ-256 dataset from Layer $64^2$ to $256^2$ and denote the number of content tokens in these layers as $(m_{64}, m_{128}, m_{256})$. The results in the following table show that more content tokens could provide more fine-grained control over the whole images, leading to better results in image generation.
>
> | $(m_{64}, m_{128}, m_{256})$ | FID |
> | :-: | :-: |
> |$(16^2, 16^2, 32^2)$  | 18.54 $\pm$  0.087|
> |$(32^2, 32^2, 64^2)$  | 15.69 $\pm$  0.067 |
> |$(64^2, 64^2, 128^2)$ | **6.81 $\pm$ 0.044** |
>
> 2) **Style mixing**. To control the coarse/middle/fine styles at different layers, we have followed StyleGANs and adopted the style mixing technique at different layers (P6 Line 182). Furthermore, we applied the style mixing technique within each layer to prevent the network from assuming that a set of style tokens are correlated (P6 Line 183-185). We will incorporate the following ablation studies on the applying of style mixing within each layer in the final version.
>
> | mixing ratio| FID |
> | :-: | :-: |
> |0%  | 6.93 $\pm$ 0.042 |
> |50% | 6.86 $\pm$ 0.064 |
> |90% | **6.81 $\pm$ 0.044** |
> |100%| 7.23 $\pm$ 0.058 |
>
> The table shows FIDs for networks trained by enabling the mixing technique for different percentages of training examples. The results show that a percentage of applying style mixing at 90% shows the best results. We apply the style mixing technique in 90% of training examples in the final model.
>
> 3) **Style modulation.** Since StyleGAN2 has verified the superiority of using weight normalization (amplifying operation) over the adaptive normalization used in StyleGAN, we chose to conduct the ablation study on different normalization methods in Table 4 and found that LayerNorm performs the best in our experiments.
>
>
> **4. Complexity and efficiency.** It takes about 52 hours to train TokenGAN with eight 16G NVIDIA V100 on FFHQ-256, in comparison with StyleGAN2's 46 hours. We report the inference time of generating an image by TokenGAN and StyleGAN2 in the table below. Specifically, we test on a single NVIDIA V100 GPU 100 times and report their average with error bars.
>
> | Model | resolution | inference time (ms) |
> | :-: | :-: | :-: |
> | StyleGAN2   | $1024^2$ | 22.33 $\pm$ 0.17
> | TokenGAN    | $1024^2$ | 26.35 $\pm$ 0.09
> | StyleGAN2   | $256^2$ | 15.83 $\pm$  0.04
> | TokenGAN    | $256^2$ | 16.10 $\pm$  0.07
>
>
> **5. Visualization of style mixing.**
> We report the visualization of style mixing in [**[this link to the results on FFHQ]**](https://www.dropbox.com/s/y1jkkqjkuc6tr19/style_editing_v0.png?dl=0). All the results are obtained by editing the style token of interest and then re-synthesizing images using edited style tokens. We provide analysis of the style tokens from the visualization as following.
>
> 1) TokenGAN controls coarse/middle/fine styles by the style tokens at different layers. Specifically,  the style tokens at **low-resolution layers ($4^2-8^2$) control coarse styles** (e.g., pose in the first row), and the style tokens at **middle-resolution layers ($16^2-64^2$) control middle styles** (e.g., hair length in the second row) and the style tokens at **high-resolution layers ($128^2-256^2$) control fine styles** (e.g., color scheme in the last row).
>
> 2) TokenGAN is able to change the pose of generated faces in a **global** way while keeping other styles unchanged (e.g., the background in the first row). Besides, TokenGAN is able to **locally** edit the hair length in the second row while remaining other regions.
>
> 3) Compared with [**[the results of the state-of-the-art (StyleGAN) in this link]**](https://drive.google.com/file/d/1U-nlMDtpnf1RcYkaFQtbh5oxnhA97hy6/view?usp=sharing), TokenGAN shows better style controllability of the generated images. For example, TokenGAN is able to change the pose of faces in the first row without changing the gender, while StyleGAN tends to change both the pose and the gender of the faces in the second row of their results.
>
>
> **6. Implementation details.** We will incorporate the following details in the final version per your suggestions.
>
> 1) We follow StyleGAN2 and adopt a skip-generator architecture in TokenGAN. Such a skip generator generates images in each layer at different resolutions (e.g., from $4^2$ to $1024^2$) and progressively upsamples and sums the images from the previous layer by skip connections to the next layer. After progressive summing, the generator takes the output in the last layer as the final results. In each layer of TokenGAN, we decode the visual tokens of images from the sequence of style-modulated content tokens, then pack them together and reshape them to a 2D shape as the images. Such a design helps TokenGAN to control coarse/middle/fine styles at low-/middle-/high-resolution layers.
>
> 2) Since TokenGAN conducts the style modulation by a cross-attention mechanism, the global consistency over all the regions in the image can be ensured by assigning similar styles to similar regions (e.g., the same color for two eyes). We further apply low-pass filtering in each layer on decoded tokens to fuse neighboring context information and ensure local consistency (R. Zhang, Making Convolutional Networks Shift-Invariant Again, in ICML 2019)
>
> **7. Discussions with [21].** Since [21] has not been published before our submission, we didn't include further discussions and comparisons in our paper. Per your suggestions, we will incorporate the following discussions in the final version. The core differences come from 1) different motivations and 2) different image resolutions.
>
> 1) Different motivations. [21] focuses on introducing bipartite structure to maintain computation of linear efficiency in transformer, while TokenGAN aims at introducing a new perspective, i.e., a token-based structure for image generation and applying attention-based style modulation for the tokens, leading to finer-grained controllability of the styles.
>
> 2) Image resolution. [21] is limited to generating images at $256\times256$ resolution due to the bipartite structure, while TokenGAN is able to generate images up to $1024\times1024$ resolution.

---

> > ### Comment · Reviewer_cPNP · 2021-09-02
> > **After rebuttal**
> >
> > The authors addressed my questions during the rebuttal. So I raised my score from borderline reject to borderline accept. It incorporates transformers into the StyleGAN structure, and achieves comparable performance with StyleGAN. In the rebuttal the authors also show that TokenGAN achieves better style controllability of the generated images, which verifies the motivation that the token-based generator enables a local content-dependent manipulation of each visual token, leading to fine-grained style learning for image synthesis. The limitation is that the combination of styleGAN and transformer is quite straightforward, and the visual quality of the generated images does not show much improvement upon StyleGAN.

---

### Official Review · Reviewer_skfS · 2021-07-16

**Rating:** 6
**Confidence:** 3

**Summary:**

This paper proposes to adopt the StyleGAN architecture (i.e. individual styles that affect image generation) to a transformer based model for image generation. The model generates individual image patches which are modulated by style tokens through an attention mechanism. The model is evaluated on the FFHQ and LSUN Church datasets and performs similarly well or better than StyleGAN2 based on FID, precision, and recall.

**Limitations And Societal Impact:**

Limitations and societal impact are not discussed.

**Main Review:**

The paper proposes to model images as a sequence of image patches. These image patches start out as learned constants (similarly to StyleGAN) and are then modulated via style tokens. The interaction between content and style tokens is modeled via an attention mechanism (between content and style) such that this approach also scales to high-resolution images (1024px).

It would be helpful to be a bit more clear about the dimensions of the individual tokens, i.e. what values do you use for mn, n, p, and d in Fig 2. Also, how many layers do you use? Since you compare to StyleGAN2 it would also be helpful to mention training time and other requirements (e.g. memory) compared to StyleGAN2, as well as how long it takes to generate a new image at different resolutions.

Since you also use stlye mixing I would like to see some qualitative results of that, i.e. what do generated images look like that are generated from two different styles? Can we get similar behavior as in StyleGAN where we can control coarse layout and texture details at different layers? Overall I would be happy to see a more detailed analysis of the "style space" in this approach, given that this is arguably one of the biggest advantages of the original StyleGAN architecture.

Since you use different styles to directly modulate individual content tokens (i.e. image patches): can you get localized behaviors for the style tokens, e.g. mix two styles "spatially" where e.g. one half of the image is controlled by one style and the other half by another style? Or do reference based image translation, e.g. replace the hair in a given image with the hair of another image?

**Time Spent Reviewing:**

3

---

> ### Author Response · Authors · 2021-08-06
> **Experimental details and results of style mixing and local editing**
>
>
> We sincerely appreciate the reviewer’s time, effort, and expertise. We address major concerns as below.
>
> **1. Experimental details**. We will incorporate the following details in the final version.
>
> 1) We use $n=32$ style tokens in the final model and the number of content tokens $m$ is growing progressively with layers for different resolutions (from $8^2$ to $1024^2$). Specifically, with $1024^2$ output resolution, the token-based generator contains a total of 15 layers of style blocks, where the first one corresponds to $8^2$ resolution, the next two correspond to $16^2$, and so forth. We empirically set the dimensions of the content tokens  $(p, m, d)$ in different layers according to the image size for the best performance and training efficiency. The details are listed below, which are also specified in our submitted codes.
>
> | spatial size | p | m | d |
> | :-: | :-: | :-: | :-: |
> | $8^2$   | 1 | $(8/1)^2$  | 512
> | $16^2$  | 1 | $(16/1)^2$ | 512
> | $32^2$  | 1 | $(32/1)^2$ | 512
> | $64^2$  | 1 | $(64/1)^2$ | 256
> | $128^2$ | 2 | $(128/2)^2$ |128
> | $256^2$ | 2 | $(256/2)^2$ |64
> | $512^2$ | 4 | $(512/4)^2$ |32
> | $1024^2$ |4 | $(1024/4)^2$ |16
>
>  2) It takes about 52 hours to train TokenGAN with eight 16G NVIDIA V100 on FFHQ-256 [26], in comparison with StyleGAN2's 46 hours. We report the inference time of generating a new image by TokenGAN and StyleGAN2 in the table below. Specifically, we test on a single NVIDIA V100 GPU 100 times and report their average and error bars.
>
> | Model | resolution | inference time (ms) |
> | :-: | :-: | :-: |
> | StyleGAN2   | $1024^2$ | 22.33 $\pm$ 0.17
> | TokenGAN    | $1024^2$ | 26.35 $\pm$ 0.09
> | StyleGAN2   | $256^2$  | 15.83 $\pm$ 0.04
> | TokenGAN    | $256^2$  | 16.10 $\pm$ 0.07
>
> **2. Application of style mixing.** Yes, as shown in [**[this link to the results on FFHQ256]**](https://www.dropbox.com/s/y1jkkqjkuc6tr19/style_editing_v0.png?dl=0), TokenGAN has a similar behavior as in StyleGAN2, i.e., controlling coarse/middle/fine styles at different layers. All the results are obtained by editing the style token of interest and then re-synthesizing images using edited style tokens.
>
> 1) The style tokens in the **low-resolution layers ($4^2-8^2$) controls coarse styles (e.g., identity and pose)**. For example, in the first row, editing a style token for pose turns the face from the side to the front.
>
> 2) The style tokens in the **middle-resolution layers ($16^2-64^2$) control middle styles (e.g., hair, beards, and mouth)**. For example, in the 2nd and 3rd row, editing a style token for the hair is able to change the hairstyle.
>
> 3) The style tokens in the **high-resolution layers ($128^2-256^2$) controls fine styles (e.g., color scheme)**. For example, in the 6th row, editing a style token for the color scheme can change the color of clothes in images.
>
> Per your suggestions, we will incorporate the results of style mixing in the final version.
>
>
> **3. Application of local editing.**
> Yes, as shown in  [**[this link to the results on FFHQ256]**](https://www.dropbox.com/s/y1jkkqjkuc6tr19/style_editing_v0.png?dl=0), TokenGAN is able to get localized behaviors for the style tokens and it can "spatially" control the styles in a global or a local way. All the results are obtained by editing the style token of interest and then re-synthesizing images using edited style tokens.
>
> 1) Take the first row as an example, editing the style token for the pose is able to **"globally" change the pose of face** while keeping the styles of other regions (e.g., hairstyle, background) unchanged.
>
> 2) Take the second row as an example, editing the style token for hair length, TokenGAN **"locally" turns the hair longer** while remaining the styles of other regions (e.g., pose and mouth).
>
> Per your suggestions, we will incorporate the results in the final version.

---

### Author Response · Authors · 2021-08-10
**To all reviewers**

We first thank all reviewers for their time reviewing this paper.

In this paper, we introduce a new perspective of image generation by **using a token-based generator (TokenGAN)**. Under this new perspective, we naturally introduce the style tokens, the content tokens, and the Transformer-based content-aware style modulation. Both quantitative (Table 2) and qualitative results (Figure 5) have verified the effectiveness of our approach.

In this rebuttal, according to the reviewers' suggestion, we report the results of image editing in [**[this link to the results of TokenGAN on FFHQ]**](https://www.dropbox.com/s/y1jkkqjkuc6tr19/style_editing_v0.png?dl=0). All the results are obtained by editing the style token of interest and then re-synthesizing images using edited style tokens. Compared with [**[the results of the state-of-the-art (StyleGAN) in this link]**](https://drive.google.com/file/d/1U-nlMDtpnf1RcYkaFQtbh5oxnhA97hy6/view?usp=sharing), TokenGAN shows significantly better style controllability of the generated images. For example, TokenGAN is able to change the pose of faces in the first row without changing the gender, while StyleGAN tends to change both the pose and the gender of the faces in the second row of their results.

Many reviewers recognized the novelty and the advantages of TokenGAN, and we appreciate these positive comments. We believe that this new perspective would inspire others and make a contribution to the community.

---

### Decision · Program_Chairs · 2021-09-27

**Decision:**

Accept (Poster)

**Comment:**

The paper proposed a transformer-based generator architecture for GAN and showed it achieved comparable image synthesis results with the StyleGAN2 baseline. It initially received a mixed rating, with two reviewers rated it above the bar and two below the bar. The rebuttal addressed some of the concerns. Both of the reviewers that were originally leaning toward a rejection recommendation upgraded the score. Overall, the meta-reviewer agreed that the proposed method has its merit. While, given the success of transformers for various vision tasks, it was kind of expected that one could achieve comparable results to CNNs with a transformer-based architecture for a GAN generator, it is still nice to see such a design being revealed and shown comparable performance. While it is unfortunate that the discriminator is still based on CNNs and the additional capability achieved by the architecture is underwhelming, the meta-reviewer thought the paper still presents a good first step. The authors are encouraged to incorporate the reviewer feedback in the final version.